# Correlation of Dengue and Meteorological Factors in Bangladesh: A Public Health Concern

**DOI:** 10.3390/ijerph20065152

**Published:** 2023-03-15

**Authors:** Md. Aminul Islam, Mohammad Nayeem Hasan, Ananda Tiwari, Md. Abdul Wahid Raju, Fateha Jannat, Sarawut Sangkham, Mahaad Issa Shammas, Prabhakar Sharma, Prosun Bhattacharya, Manish Kumar

**Affiliations:** 1COVID-19 Diagnostic Lab, Department of Microbiology, Noakhali Science and Technology University, Noakhali 3814, Bangladesh; 2Advanced Molecular Lab, Department of Microbiology, President Abdul Hamid Medical College, Karimganj 2310, Bangladesh; 3Department of Statistics, Shahjalal University of Science and Technology, Sylhet 3114, Bangladesh; 4Department of Health Security, Expert Microbiology Research Unit, Finnish Institute for Health and Welfare, 70701 Kuopio, Finland; 5Department of Public Health, North East University, Sylhet 3100, Bangladesh; 6Department of Environmental Health, School of Public Health, University of Phayao, Muang District, Phayao 56000, Thailand; 7Department of Civil and Environmental Engineering, College of Engineering, Dhofar University, P.O. Box 2509, Salalah PC 211, Oman; 8School of Ecology and Environment Studies, Nalanda University, Rajgir 803116, India; 9COVID-19 Research, Department of Sustainable Development, Environmental Science and Engineering, KTH Royal Institute of Technology, Teknikringen 10B, SE 10044 Stockholm, Sweden; 10Sustainability Cluster, University of Petroleum and Energy Studies, Dehradun 248007, India; 11Escuela de Ingeniería y Ciencias, Tecnologico de Monterrey, Campus Monterey, Eugenio Garza Sada 2501 Sur, Monterrey 64849, Mexico

**Keywords:** Dengue virus (DENV), Dengue outbreak, Dengue fever, meteorological factors, Dengue cases and deaths, time series models, Bangladesh

## Abstract

Dengue virus (DENV) is an enveloped, single-stranded RNA virus, a member of the *Flaviviridae* family (which causes Dengue fever), and an arthropod-transmitted human viral infection. Bangladesh is well known for having some of Asia’s most vulnerable Dengue outbreaks, with climate change, its location, and it’s dense population serving as the main contributors. For speculation about DENV outbreak characteristics, it is crucial to determine how meteorological factors correlate with the number of cases. This study used five time series models to observe the trend and forecast Dengue cases. Current data-based research has also applied four statistical models to test the relationship between Dengue-positive cases and meteorological parameters. Datasets were used from NASA for meteorological parameters, and daily DENV cases were obtained from the Directorate General of Health Service (DGHS) open-access websites. During the study period, the mean of DENV cases was 882.26 ± 3993.18, ranging between a minimum of 0 to a maximum of 52,636 daily confirmed cases. The Spearman’s rank correlation coefficient between climatic variables and Dengue incidence indicated that no substantial relationship exists between daily Dengue cases and wind speed, temperature, and surface pressure (Spearman’s rho; r = −0.007, *p* > 0.05; r = 0.085, *p* > 0.05; and r = −0.086, *p* > 0.05, respectively). Still, a significant relationship exists between daily Dengue cases and dew point, relative humidity, and rainfall (r = 0.158, *p* < 0.05; r = 0.175, *p* < 0.05; and r = 0.138, *p* < 0.05, respectively). Using the ARIMAX and GA models, the relationship for Dengue cases with wind speed is −666.50 [95% CI: −1711.86 to 378.86] and −953.05 [−2403.46 to 497.36], respectively. A similar negative relation between Dengue cases and wind speed was also determined in the GLM model (IRR = 0.98). Dew point and surface pressure also represented a negative correlation in both ARIMAX and GA models, respectively, but the GLM model showed a positive association. Additionally, temperature and relative humidity showed a positive correlation with Dengue cases (105.71 and 57.39, respectively, in the ARIMAX, 633.86, and 200.03 in the GA model). In contrast, both temperature and relative humidity showed negative relation with Dengue cases in the GLM model. In the Poisson regression model, windspeed has a substantial significant negative connection with Dengue cases in all seasons. Temperature and rainfall are significantly and positively associated with Dengue cases in all seasons. The association between meteorological factors and recent outbreak data is the first study where we are aware of the use of maximum time series models in Bangladesh. Taking comprehensive measures against DENV outbreaks in the future can be possible through these findings, which can help fellow researchers and policymakers.

## 1. Introduction

Dengue is caused by RNA Dengue virus (DENV), which contains ~11 kb RNA genome including an ORF (open reading frame), three structural proteins, namely capsid (C), pre-membrane or membrane (prM/M), an envelope (E) [1]. Aedes female mosquito species carrying one of the four serotypes, such as DEN-1, DEN-2, DEN-3, and DEN-4, are the carriers of the acute febrile viral illness Dengue fever. Over 390 million individuals worldwide suffer from this illness each year, and 50% of the world’s population belongs to the risk group [2]. The gold standard RT-PCR can find viral titers that aid patient diagnosis and therapy planning [3,4,5]. The clinical manifestations of Dengue infection are symptomatic, asymptomatic, or mild illness (such as fever, headache, myalgia, decreased platelet counts, and leucopenia), similar to flu symptoms [3]. Dengue hemorrhagic fever (DHF), Dengue shock syndrome (DSS), and life-threatening scenarios are sometimes considered in the case of severe illnesses [6]. However, DHF hematomas are found when patients are characterized by thrombocytopenia or extremely low platelet counts [1].

From an analysis of previous data, it was obtained that the climatic factors, especially temperature, rainfall, and humidity, have abruptly changed in the past few years [7]. The Dengue outbreaks and positive cases also accrued with the discrepancy of the climate. Meteorological conditions are linked to Dengue disease because *Aedes Aegypti* growth and life cycle are affected by rainfall, temperature, humidity, and wind, either directly or indirectly [7]. It has been reported that rising temperatures and precipitation have aided in the rise of Dengue incidences [8]. Such settings are advantageous for breeding places that promote vector growth as a result of an increase in human and vector interaction that enhances viral transmission from an infected person to a new person. These environmental modifications may flare up the genetic mutation of the viruses [8]. According to Mutsuddy et al. (2019), climate alteration may affect some vectors for multiplication or extinction [9]. In 70% of Asian nations, Dengue infection is a severe public health concern [4]. The geographic location, climatic factors, rapid urbanization, deforestation, water pollution, lack of wastewater management system, high population density, and ineffective vector control strategies are responsible for DENV outbreaks. Several previously published articles reported relationships between various meteorological factors and Dengue cases, but this relationship has not been tested in recent years. Using data from 2000 to 2023 and five significant time series models, we thoroughly examined the association between meteorological factors and Dengue cases in this study. This study can guide fellow researchers and policymakers in tackling the Dengue outbreak.

## 2. Materials and Methods

### 2.1. Dengue Cases and Meteorological Factors

Confirmed positive cases were downloaded from the Directorate General of Health Services (DGHS)’s website for patients from all over Bangladesh taking eight divisions (Appendix A). In this study, the daily DENV new cases between 1 January 2000 and 31 January 2023 were used from the DGHS website (https://old.dghs.gov.bd/index.php/bd/home/5200-daily-dengue-status-report) (accessed on 10 January 2023). We also obtained meteorological information from the NASA website to perform a time series analysis (NASA, 2022) (Appendix A) (https://www.nasa.gov/) (accessed on 10 January 2023). In this study, we considered daily dew point (°C), daily temperatures (°C), precipitation (mm), relative humidity (%), surface pressure (kPa), and wind velocity (m/s) at a level of 2 m height above ground level (Appendix A).

### 2.2. Statistical Time Series Models

The Simple Exponential Smoothing Model (SES), Auto-Regressive Integrated Moving Average Model (ARIMA), Seasonal Auto-Regressive Integrated Moving Average Model (SARIMA), and GA time series models were used in this work to forecast recent Dengue cases. Additionally, Auto-Regressive Integrated Moving Average with Explanatory Variables (ARIMAX), Generalized Additive Model (GA), and Generalized Linear Mixed Models (GLM) time series models were used in this work to evaluate the correlations between meteorological parameters and Dengue cases. The SES model is utilized as a baseline to assess the prediction accuracy of other models in this investigation. All models were used to forecast new DENV cases for 30 days [10,11,12].

#### 2.2.1. Simple Exponential Smoothing Model (SES)

SES forecasting enables a short-term prediction to use a reasonably steady mean while assuming data changes [13,14,15] created the original version of this model, which is still a helpful observation technique, and its application has been rapidly expanding in recent years. SES was noted as one of the most well-liked and potent models ideal for investigations where there is no clear trend or weather pattern, according to Weller and Crone (2012). The R package “fpp2” created the SES model, a univariate time series prediction. The following equation can present the SES model:Vt = V _(t−1)_ + α(v _(t−1)_ – V _(t−1)_)(1)

In Equation (1), vt denotes the actual value of the series at time t, Vt stipulates the forecast value of the series at the time as well as t, and α is a weighting factor that takes a value between 0 and 1.

In Equation (1), vt stands for the series’ actual value at time t, Vt for its forecasted value at the desired time, and α is a weighting factor that can take a value between 0 and 1.

#### 2.2.2. Auto-Regressive Integrated Moving Average Model (ARIMA)

A statistical time series analysis technique called the auto-regressive integrated moving average model (ARIMA) uses time series data to analyze data sets and forecast trends. A time series model called the ARIMA auto-regressive integrated moving average is used in statistical analysis to analyze seasonal patterns and estimate results. This model has three steps: identification, factor prediction, and model diagnosis. Using autocorrelation (ACF) and partial autocorrelation functions, auto-regressive and shifting average constituents are chosen after seasonality and stationary recognition. The ARIMA examines opposite goals in the lack of seasonality in the time series data [16] where the “forecast R package” is used in this current study, and Equation (2) is presented below [17]:Vt = α + φ1V _(t−1)_ + … +φpV _(t−*p*)_ + θ_1e (t−1)_ + … +θqe (t − q) + Wt(2)

In this equation, α is the constant value, φ1 − φ*p* means auto-regressive model parameters, θ1 − θq indicates moving-average model parameters, and Vt for its forecasted value at the desired time, where the wt is white noise.

#### 2.2.3. Auto-Regressive Integrated Moving Average with Explanatory Variables (ARIMAX)

Auto-Regressive Integrated Moving Average with Explanatory Variables (ARIMAX) is another significant and familiar time series model which allows external parameters, such as climatic factors, to upgrade the analysis and forecasting accuracy [18,19,20,21]. This model analyzes the association between the time series data for arranging genuine designs, exploiting the straight perceptions, and deleting high-frequency commotion.

#### 2.2.4. Seasonal Auto-Regressive Integrated Moving Average Model (SARIMA)

Seasonal-ARIMA, or SARIMA, is another time series model analogous to ARIMA but more precise and used for forecasting trends and seasonal alterations. These models are used for contrasting the time series data for seasonal frequency, although for non-seasonal cases [22,23,24,25].

#### 2.2.5. Generalized Additive Model (GA)

The GAM, or generalized additive model, is applied for analyzing the interaction of climatic factors with Dengue cases already being used for various research purposes. This time series model is more eminent and user-friendly, which is preferable for analyzing national morbidity, mortality, and air pollution studies. GAM fits the generalized additive model for parametric and nonparametric regression and smoothing. It also augments the conventional GLM (generalized linear models) by changing linear predictors of the form η = Σj βjxj with η = Σjfj (xj). fj (xj) is used here for a nonparametric function R package for analyzing GAM in this study.

#### 2.2.6. Generalized Linear Mixed Models (GLM)

The GLMM, or Generalized Linear Mixed Model, is a statistical model that augments the Generalized Linear Model (GLM). This model is used to analyze the random effects from clustered categorical data. The ability of the GLMM model to distinguish the impact of nested data is a significant advantage. The season is the second category in the current study, while year characteristics are used as repeated observations. A standard Equation (3) of this model is presented below:y = Xβ + Zu + ε(3)

This equation denotes anN × 1 as the outcome variable; X is anN × *p* matrix; β is a *p* × 1 column vector; N × q is the design matrix; u is a q × 1 vector of the random; and ε is anN × 1 column vector of the residuals.

### 2.3. Statistical Analysis

This study applied five statistical time series models to see the trend and forecasting of Dengue cases for the recent future, along with forecasting from three time series models showing a correlation between Dengue cases and meteorological conditions. Additionally, Spearman’s rank correlation coefficients were employed in this investigation. For determining seasonal variation, we used the Poisson regression model in three major seasons in Bangladesh. Data can be characterized by a Poisson distribution when observations are counted in whole numbers and when event occurrences are independent (one event occurrence does not affect the chance of another event occurring), and when the specifics of the observed time interval are known and are the same for each participant.

## 3. Results

### 3.1. Dengue Status in Bangladesh

In 2022, the first Dengue case was confirmed in the month of January, and 126 Dengue-positive patients were recorded. From the DGSH data collection, it can be shown that Dengue cases rose from April 2022 to November (19,334) and December (5024), including May (163 cases), June (737), July (1571), August (3521), September (9911), and October (21,932) (Figure 1). The highest number of mortality was recorded in November (113) and October (86). The Dengue outbreak was tremendously devastating, with 62,382 patients in 2022 compared with only 28,429 confirmed cases in the year 2021 (Figure 1).

The mean for Dengue cases is 882.26, with a standard deviation (SD) of 3993.18, a minimum of 0, and a maximum of 52,636 (Table 1). Table 1 also points out that there are 52,636 maximum cases in Bangladesh. The average daily confirmed cases are approximately 882.26, while daily average values for temperature, dew point, relative humidity, precipitation, surface pressure, and wind speed are 25.38 °C, 19.80 °C, 74.78%, 5.81 mm/day, 100.68 kPa, and 1.93 m/s, respectively (Table 1). The descriptive analysis of this study stipulates that the lowest temperature is 15.39 °C, whereas the highest is 31.86 °C from 1 January 2000 to 31 January 2023 in Bangladesh. In addition, precipitation is found to be a minimum of 0 mm/day and a maximum of 28.39 mm/day. Moreover, the lowest surface pressure is recorded as 99.75 kPa, and the highest is 101.52 kPa.

### 3.2. Association between Daily Dengue Cases and Meteorological Variables

Correlations between meteorological parameters and Dengue cases are presented in Figure 2. This graph suggests that none of the associations are statistically significant. However, Spearman’s rank correlation coefficients between meteorological variables and confirmed daily Dengue cases indicate a meaningful but weak relationship between Dengue cases and climatic parameters (Figure 2). Concerning daily Dengue cases, the mean temperature and wind speed show a weakly negative link (r = −0.007, *p* > 0.05 and r = 0.085, *p* > 0.05, respectively). However, there is a significant positive correlation among dew point, relative humidity, and rainfall with daily Dengue cases (r = 0.158, *p* < 0.05, r = 0.175, *p* < 0.05, and r = 0.138. *p* < 0.05), respectively. On the contrary, surface pressure exhibits an insignificant negative correlation with daily Dengue cases (r = −0.086, *p* > 0.05) (Figure 2).

### 3.3. Time Series Model Results

The forecasting results from the time series models are displayed in Figure 3, along with the confirmed and projected Dengue cases from January 2000 to January 2023. With AIC, AICc, and BIC of 6083.90, 6083.99, and 6094.77, respectively, we describe a consistent pattern between observed and predicted national Dengue cases in the SES model (Table 2 and Figure 3).

Dengue with AIC, AICc, and BIC of 5232.86, 5233.01, and 5247.34 for the ARIMA model, compared with 5088.54, 5088.77, and 5106.44 for the SARIMA model, and 5238.35, 5239.18, and 5274.56 for the ARIMAX model, we detected a strong, growing trend between observed and predictive Dengue cases. However, the GA model performs weakly (i.e., AIC = 5358.541, AICc = 5358.870, and BIC = 5396.837). All models, except for SES, forecast a considerable increase in cases of Dengue over the following 30 days (Figure 3).

In the ARIMAX and GA models, wind speed (−666.50 [95% CI: −1711.86 to 378.86] and −953.05 [−2403.46 to 497.36], respectively) and dew point (−102.00 [−1308.42 to 1104.43] and −616.31 [−2079.18 to 846.55], respectively) have a slight negative interaction with Dengue cases. Nevertheless, temperature and relative humidity are positively affiliated with Dengue cases (105.71 [−1034.44 to 1245.87] and 633.86 [−2079.18 to 846.55], respectively) in the ARIMAX model, and the GA model also exhibits identical results. In contrast, rainfall is negatively associated with the ARIMAX model (−69.52 [−178.61 to 39.58]) and positively associated (21.59 [−114.83 to 158.02]) in the GA model (Table 3).

In the GLM model, temperature and dew point (0.55 [95% CI: 0.31 to 0.98] and 2.06 [1.11 to 3.82], respectively) have a significant association with Dengue cases. Temperature displayed a negative association, whereas dew point stipulates a positive association according to the GA model. However, wind speed, relative humidity, rainfall, and surface pressure are fruitlessly associated with Dengue cases, though wind speed and rainfall (0.98 [95% CI: 0.68 to 1.42] and 1.01 [0.98 to 1.04], respectively) have a positive association with Dengue cases. Contrarily, relative humidity (0.88 [95% CI: 0.75 to 1.04]) negatively affects Dengue cases (Table 3).

In the Poisson regression model, temperature and relative humidity (1.30 [95% CI: 1.29 to 1.31] and 1.15 [1.14 to 1.15], respectively) have a substantial positive correspondence with Dengue cases in the winter season. In that season, wind speed and rainfall (0.01 [95% CI: 0.01 to 0.02] and 1.08 [1.07 to 1.09], respectively) have a substantial negative correspondence with Dengue cases. In the summer, temperature (IRR = 1.39), dew point (6.37), rainfall (1.08), and surface pressure (10.59) have a significant positive association with Dengue cases (Table 4). In that season, wind speed (0.65) and relative humidity (0.84) have a significant antagonistic association with Dengue cases. In the monsoon season, temperature (2.52), rainfall (1.08), and surface pressure (10.57) have an essential curvilinear correlation with Dengue cases. In that season, wind speed (0.84) and relative humidity (0.97) significantly negatively affect Dengue cases.

## 4. Discussion

Bangladesh is still fighting against the most devastating COVID-19 outbreak, and the recent dreadful flood also struck it [26,27,28,29]. Although the SARS-CoV-2 pandemic raises awareness of zoonotic infections and the need for new vaccinations against the novel, emerging, or re-emerging viruses, other zoonotic viral diseases are also rising day by day without control [14,20,22,23]. This study hypothesizes that daily Dengue cases can be related to meteorological factors. This study has indicated that dew point, relative humidity, and rainfalls are three meteorological parameters that might be linked to daily Dengue cases. Everyday Dengue illness occurrences in Bangladesh were correlated with dew point, relative humidity, and precipitation, according to an analysis of the metrological parameters (r = 0.158, *p* < 0.05; r = 0.175, *p* < 0.05; and r = 0.138, *p* < 0.05, respectively). Other elements, such as stagnant water and sewage effluent, which also generated distinct habitats for mosquito reproduction, are associated with the rise of Dengue. Additionally, the utilization of metrological data series for more than 20 years demonstrated the connection between current, widespread sickness in Bangladesh and climate change. Another linkage could be established during the flooding period as a reduction in disease outbreaks because the flood could wash away the mosquito larvae from wetlands, marshlands, and floodplains.

In this study, a one-degree increase in dew point might increase the risk of Dengue cases two times according to the GLM model, and in the summer, it is five times according to the Poisson regression model. The GLM and Poisson models’ results revealed a substantial correlation between them for dew point and Dengue incidences. Analogous observations of a significant relationship between dew point and Dengue incidence were observed in Brazil [30]. This study also reveals that the high humidity ameliorates the Dengue cases in the winter and amortizes in the summer and monsoon seasons. High humidity favors the elevated longevity of adult mosquitoes and the shortening of the viral incubation period, thereby allowing an increased transmission intensity [31]. Humidity also affects adult mosquitoes’ survival and biting frequency [32]. The results of a few other researchers have also been inconsistent and inconclusive. Relative humidity, with a 3–4 month lag period, was shown to be the most important predicting factor for Dengue outbreaks in Indonesia, according to a study conducted there [33]. According to another study, a Dengue outbreak typically occurs the following year when there is a low relative humidity level in September and October [34].

Rainfalls escalate the water stagnation and proliferation of mosquitoes so that it may intensify DENV. In summer and monsoon seasons, a 1 mm increase in rainfall can increase Dengue cases by 8%. Rainfall-induced increases in vector density and a corresponding decline in Dengue incidence have been documented. Despite Bangladesh having a year-round potential for Dengue transmission, there have been very few winter/dry season. Dengue cases reported over the past 20 years due to a lack of moisture needed to refill frequent mosquito breeding sites [35].

A long-term study of meteorological factors with the number of Dengue cases using Poisson regression showed that the seasonal meteorological factors were positively correlated with them. This is congruent with research on the correlation between Dengue cases and climatic variables in Jeddah, Saudi Arabia, including relative humidity and temperature [36]. The climate significantly influences the prevalence of infection with Dengue hemorrhagic fever. The climate significantly impacts the frequency of Dengue hemorrhagic fever infection in the Kolaka area, according to a study by Tosepu et al. (2018). It was the country’s first study, as far as we know. However, other sources have also reported on a season-specific pattern of Dengue cases in Southeast Asia [37]. Our findings concur with other extensive research from the same geographic area, Myanmar and India [38], and the studies from the Gulf of Thailand and Puerto Rico [39]. Temperature and rainfall were discovered to be important contributing factors in earlier investigations [40]. Moreover, earlier studies presupposed that the impacts of climate factors are not seasonal. The effects of climate variables might, however, also vary throughout time. There are different ways that rainfall can affect the incidence of Dengue, according to several studies.

Additionally, earlier research presupposed that the impacts of climatic factors are not seasonal. The impacts of climatic factors might, however, also vary throughout time. Numerous studies have shown that rainfall’s impact on Dengue occurrence can change throughout the year [41]. Since rain can damage potential mosquito habitats, the heavy rain that falls during monsoon season is likely to have a detrimental impact on the growth of the mosquito population. On the other hand, rains throughout the winter might leave stagnant pools of water ideal for mosquito breeding.

Few human-made and natural occurrences in some regions alter precipitation, which affects mosquito dispersion and population size [42]. Forecasting the frequency and length of outbreaks may be performed using weather variables, including relative humidity, wind speed, lowest and peak temperatures, and average and highest values [36]. The overall number of hospitalizations in Bangladesh as of 12 October 2022, as reported by the DGSH, was 23,592 individuals, of whom 17,456 were hospitalized in Dhaka and 6136 elsewhere. According to the DGSH report on 12 October 2022, the total number of registered hospitalizations was 23,592 in Bangladesh, of which 17,456 were in Dhaka city, and 6136 were outside of Dhaka city. The Department of Health, GoB, reported that 12,875 DENV patients were admitted to different hospitals. At the same time, 10,017 were from the capital, and the second-highest numbers were found in the Chittagong division (2814) and the Khulna division (1082), while Barisal (787), Rajshahi (616), Mymensingh (267), Rangpur (36), and the Sylhet division (22) followed. In the year 2022, the total mortality was more than 70, where 45 people died in Dhaka city, 1 in Narayanganj district, 32 in Chattogram, and 5 in the Barishal division (Figure 1). The first 14 days of October saw at least 7500 patients, or about 32% of all hospitalized patients, and September alone saw 9911 patients, the most of any month. In addition, 3571 patients were admitted to the hospital in August, followed by 1571 patients in July, 737 patients in June, 163 patients in May, 23 patients in April, 20 patients each in February and March, and 126 patients in January 2022. In the Rohingya refugee camps (Forcibly Displaced Myanmar Nationals (FDMN)) in Cox’s Bazar district, Bangladesh, around 7687 confirmed positive DENV cases, with six deaths disclosed (case fatality rate roughly 0.08%). The replication and multiplication of vectors for this disease were highly dependent on climatic factors [28,29,43]. The climate of Bangladesh, as a part of tropical Asian countries, consists of two monsoons with year-to-year time variation. Bangladesh has a tropical-moist climate with variable seasonal rainfall, a moderately warm climate, and significant relative humidity [44]. In this country, there are typically four distinct climatic seasons per year: The winter, which lasts from December to February, is when temperatures are lowest; The summer, from March to May, is when they are highest; The rainy season, from June to September, and the fourth number, the post-monsoon autumn, which lasts from October to November [45]. From the observations of the 61 years (1960–2021) of daily temperature data, we discovered an average temperature of 25.2 °C in July, a minimum temperature of 12.9 °C in January, and a maximum temperature of 33.5 °C in April in Bangladesh. We also observed a maximum precipitation of 496 mm in July and a minimum of 4 mm in January.

It is counted that the rising world temperature may associate with escalating vector-borne diseases. According to these study findings, climatic factors are closely associated with ameliorating Dengue infection. Most positive cases in Bangladesh were recorded during the monsoon season when rainy weather promotes mosquito development and survival. The findings of this study also showed a correlation between Dengue cases and relative humidity and rainfall. The development of the Dengue virus in ambient temperature and other factors impact Aedes mosquitoes, which are daytime feeders that start to spread at about 29.3 °C. This suggested that mosquito growth is best at temperatures between 25 °C and 27 °C. Additionally, a higher temperature in the range of 27 °C to 31 °C is a good predictor for the growth of Aedes mosquitoes, having a preventative impact on the spread of Dengue. Another climatic factor, rainfall, is crucial for replicating Aedes mosquitoes, which helps lay the vector eggs [46,47,48,49].

First, appropriate preventive steps must be followed to solve the Dengue issue to avoid the affected mosquito contact. It should be mentioned that Aedes mosquitoes are more active and likely to bite during the day. As a result, people should avoid standing water and wastewater and use insecticides to kill them instead of coils and nets, long-sleeved clothing, or dresses (CDC, 2022). Vaccination and drug therapy should be developed in addition to taking different prevention steps [50,51]. Antipyretics (paracetamol) can be used for fever, analgesics, or painkillers for joint pain. Severe patients with DHF/DSS should be immediately hospitalized, where oral rehydration therapy can be followed for dehydration. Platelet transfusion is recommended when its level reaches 20,000 or below [52].

These climatic elements can be incorporated into early warning systems to improve Dengue forecasting, vector management, and control in the future [53]. Vector control should follow according to the WHO’s Integrated Vector Management (IVM) system to regulate the vector’s breeding areas, lessen the vector’s environment management, decrease the stagnant areas of water, and maximize the use of larvicides, fogging, fumigation in camps, and the treatment of waste and wastewater before releasing into the environment. Bangladesh also recently experienced the dreadful flood that commenced on 17 May 2022, and mainly captivated two districts of the northeastern division of Sylhet with 79 deaths, thousands of injuries, and enormous economic loss. Due to widespread infection, the ongoing COVID-19 infectious disease caused by SARS-CoV-2 has damaged global public health, enterprises, and economies [28,29,51]. It is not crystal clear until now why the Dengue infection is augmentation while government actions, environment, season, wastewater, and public health knowledge gap are prevalent. It can turn into another outbreak if it cannot be immediately controlled. Few environmental variables are interconnected with each other. For example, the dew point is affected by atmospheric humidity and air temperature. Humidity influences rainfall, and the quantity of rain influences water stagnation everywhere, such that it may increase the vector of DV (mosquito). Global warming, rising sea levels, and increased water stagnation in Bangladesh contribute to an increase in Dengue disease vectors.

It is almost impossible to include all of these environmental factors in current mathematical models because many works are in parallel, and many are local factors. Investigating and determining the ecological factors’ role in this outbreak can simplify interpretation. However, we believe the current study can provide a new perspective for discussion.

## 5. Conclusions 

Dengue is a flavivirus infection spread by mosquitoes and is most common in tropical and subtropical areas. The significance of meteorological factors in Dengue disease is identified in many studies where the temperature was responsible for quick viral vector replication, the humidity boosted vector transmission, and precipitation also showed different effects on the spread of mosquito eggs as well as larvae. According to this study result, climatic parameters, temperature, and relative humidity showed positive correlations with Dengue cases using 23 years of data. SARIMA scored better than the other four time series models, suggesting that we were able to estimate events more precisely when we took into account seasonal influences. These results also validate our Poisson regression model. All three models reported less correlation with Dengue cases than by the Poisson regression model.

Moreover, the Poisson model with seasonal segmented data showed a stronger correlation. This scenario is mainly attributed to inadequate education, knowledge gaps, a lack of appropriate legislative rules, and a poor sanitation system without proper wastewater management. Combining wastewater-based and serological-based hospital surveillance can detect hotspots and predict patient numbers. A multi-sectoral approach should be applied to amortize Dengue transmission, urbanization planning, environmental degradation, production of active therapeutic vaccines, and education to prevent and control the disease. Bangladesh has been attacked by Dengue many times, and one of the events recorded in 2019 is its highest case. A multifaceted approach is needed for vector control, rapid diagnosis at a low cost, a proper care system, and vaccines for four strains of Dengue. The necessity of collaborative research is crucial to gain a deeper understanding of this disease, immunity, and virus.

## Figures and Tables

**Figure 1 ijerph-20-05152-f001:**
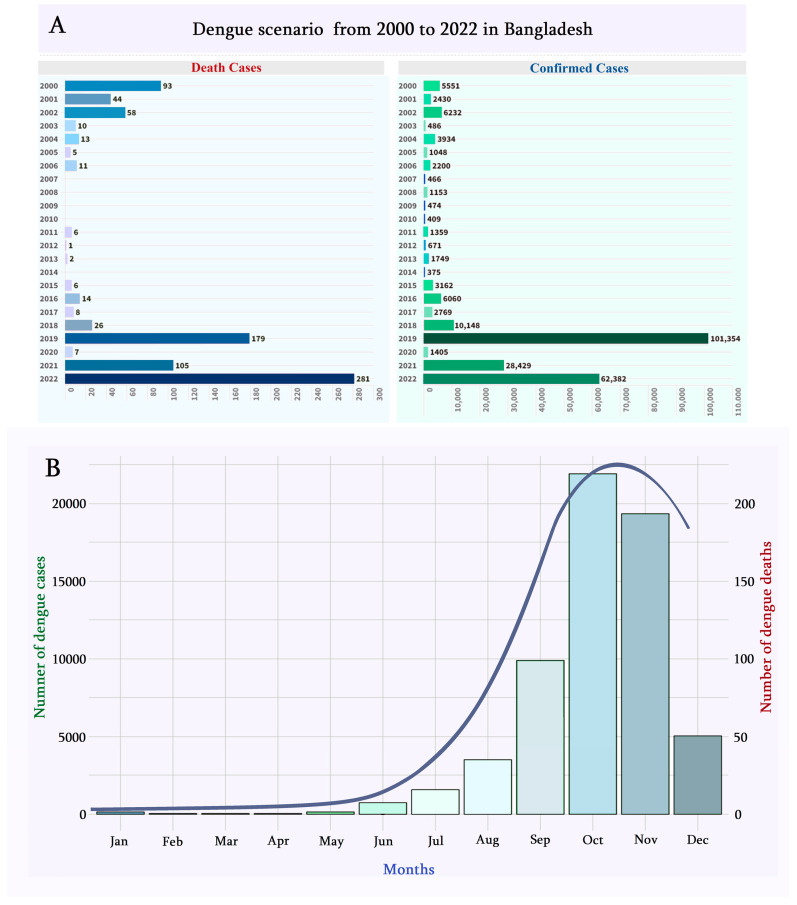
Status of Dengue patients in Bangladesh. (**A**) Number of Dengue confirmed positive and reported death cases from 2000 to 2022. (**B**) Month-wise national Dengue fever surveillance in Bangladesh, 2022.

**Figure 2 ijerph-20-05152-f002:**
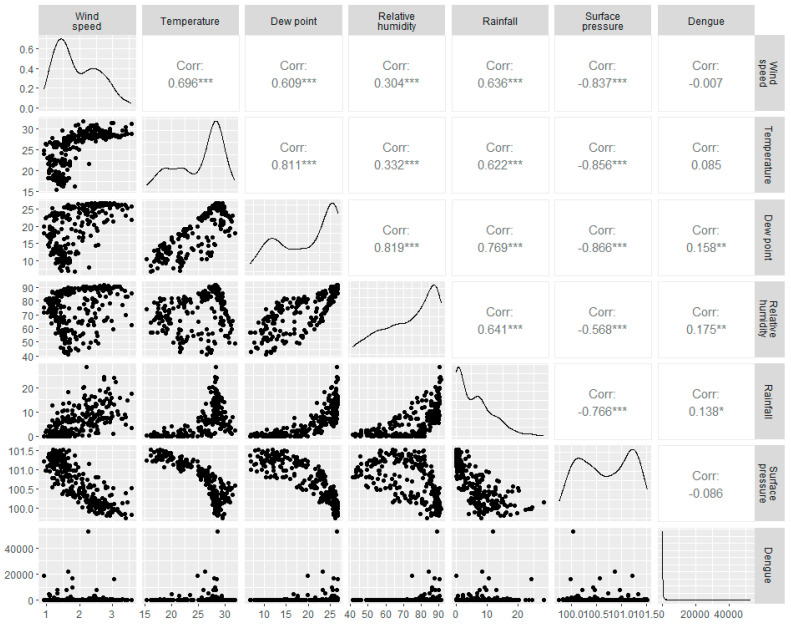
The Spearman’s rank correlation coefficient between climatic variables and Dengue incidence from 1 January 2000 to 31 January 2023 is shown in a scatter plot. (*, **, *** Significant correlation).

**Figure 3 ijerph-20-05152-f003:**
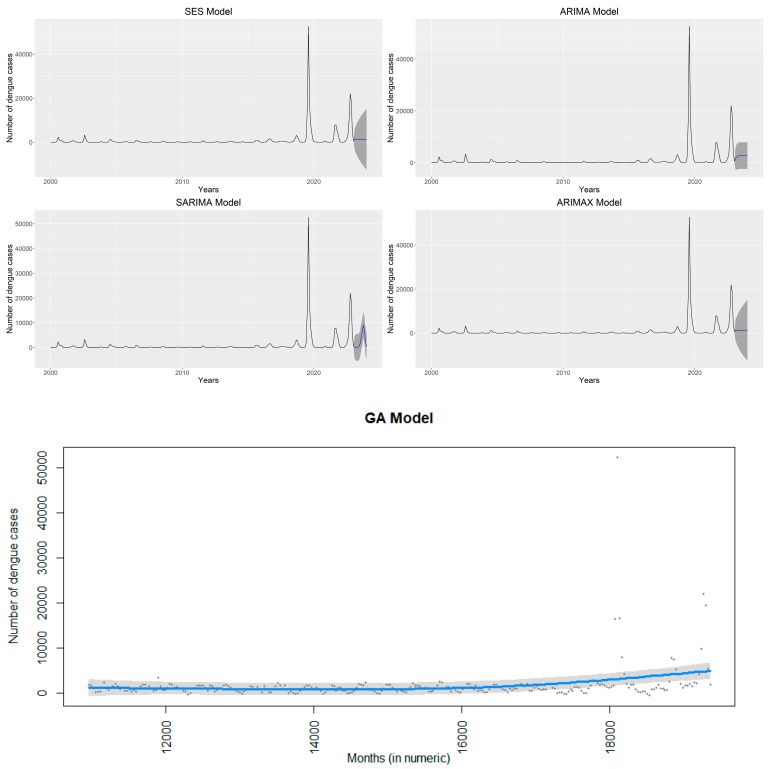
Results of time series models, the ARIMA, ARIMAX, GA, PROPHET, SARIMA, and SES models.

**Table 1 ijerph-20-05152-t001:** Summary statistics of meteorological parameters and daily Dengue confirmed cases in Bangladesh.

Variables	Mean ± SD	Minimum	Maximum
Temperature (°C)	25.38 ± 4.22	15.39	31.86
Dew/frost point temperature (°C)	19.80 ± 6.31	6.69	26.83
Relative humidity (%)	74.78 ± 14.09	41.35	91.52
Precipitation (mm/day)	5.81 ± 5.81	0	28.39
Surface pressure (kPa)	100.68 ± 0.51	99.75	101.52
Wind speed (m/s)	1.93 ± 0.64	0.91	3.58
Daily confirmed case (number of persons)	882.26 ± 3993.18	0	52636

**Table 2 ijerph-20-05152-t002:** Summary of the five time series models.

Models	SES	ARIMA	SARIMA	ARIMAX	GA
AIC	6083.90	5232.86	5088.54	5238.35	5358.54
AICc	6083.99	5233.01	5088.77	5239.18	5358.87
BIC	6094.77	5247.34	5106.44	5274.56	5396.84

**Table 3 ijerph-20-05152-t003:** Factors associated with Dengue cases using the ARIMAX and GA models.

Variables	Confirmed Cases								
	ARIMAX Model			Generalized Additive Model			GLM Model		
	Coef.	95%CI	*p*-Value	Coef.	95%CI	*p*-Value	IRR	95%CI	*p*-Value
Wind Speed	−666.50	−1711.86 to 378.86	0.211	−953.05	−2403.46 to 497.36	0.199	0.98	0.68 to 1.42	0.921
Temperature	105.71	−1034.44 to 1245.87	0.856	633.86	−2079.18 to 846.55	0.370	0.55	0.31 to 0.98	0.044
Dew Point	−102.00	−1308.42 to 1104.43	0.868	−616.31	−2079.18 to 846.55	0.410	2.06	1.11 to 3.82	0.022
Relative Humidity	57.39	−292.38 to 407.16	0.748	200.03	−215.87 to 615.92	0.347	0.88	0.75 to 1.04	0.132
Rainfall	−69.52	−178.61 to 39.58	0.212	21.59	−114.83 to 158.02	0.757	1.01	0.98 to 1.04	0.493
Surface Pressure	−1626.65	−4119.53 to 866.24	0.201	−564.60	−3690.62 to 2561.42	0.724	1.86	0.87 to 2.99	0.111

**Table 4 ijerph-20-05152-t004:** Factors associated with Dengue cases using the Poisson regression model in different seasons in Bangladesh.

Variables	Confirmed Cases								
	Poisson Regression Model: Winter			Poisson Regression Model: Summer			Poisson Regression Model: Monsoon		
	IRR	95%CI	*p*-Value	IRR	95%CI	*p*-Value	IRR	95%CI	*p*-Value
Wind Speed	0.01	0.01 to 0.02	<0.001	0.65	0.59 to 0.71	<0.001	0.84	0.82 to 0.85	<0.001
Temperature	1.30	1.29 to 1.31	<0.001	1.39	1.04 to 1.85	0.026	2.52	2.50 to 2.55	<0.001
Dew Point	-	-	-	6.37	4.77 to 8.57	<0.001	-	-	-
Relative Humidity	1.15	1.14 to 1.15	<0.001	0.84	0.78 to 0.90	<0.001	0.97	0.97 to 0.98	<0.001
Rainfall	1.08	1.07 to 1.09	<0.001	1.08	1.07 to 1.09	<0.001	1.08	1.08 to 1.09	<0.001
Surface Pressure	-	-	-	10.59	8.34 to 13.44	<0.001	10.57	10.05 to 11.11	<0.001

## Data Availability

Data in this manuscript extracted from Open access data bases and will be available for all on the basis of their request.

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
