# Peer review of "Correlation of Dengue and Meteorological Factors in Bangladesh: A Public Health Concern"

_ijerph, 2023, doi:10.3390/ijerph20065152_

Round 1
Reviewer 1 Report
In this article six statistical models are applied to test the relationship between dengue-positive cases and meteorological parameters in Bangladesh. The idea of studying the association of dengue and meteorological variables, in a period of 23 years, is interesting. However, some parts of the manuscript are confusing. Please review the writing of the manuscript.
Some sections of the manuscript (Introduction and conclusion) should be more directed to the relationship between dengue and meteorological variables. Another important point is that it is not clear the time period that was included in some of the analyses. It is explained that 23 years were covered in the study, but in some parts it is mentioned that data are from 2020 to 2022
Abstract:
Line 39: “During the study period, the mean daily DENV cases was 831.33±(3877.51), ranging between 0 to 52636”. As it is written it is understood that the maximum of daily dengue cases is 52636.
Line 39: Not clear to which model(s) the authors are referring: “Though no substantial relationship exists between daily dengue cases and temperature, wind speed, and surface pressure.”
-Why in the abstract only the results of the ARIMAX and GA models are shown?
-Line 43-45: It is mentioned the values of the ARIMAX and GA model, but is not clear if there is a relationship or not.
-Introduction:
The article is assessing the correlation between dengue and meteorological data. Complement the introduction with more information related to dengue and meteorological factors.
It is not necessary to include the information that is given between line 64 and 70.
Figure 1 is not necessary.
Results:
Line 193: It is not clear the period of time when the 126 dengue cases were recorded.
Line 198: “Although the Dengue outbreak was tremendously devastating with 101354 patients in 2022”. It is not clear how this number is obtained. Figure 2A shows that in 2022 there are 62382 cases
Line 200: “the ongoing epidemic identified 16092 positive patients”. Not clear the period of time when these cases are recorded, what do you mean by the ongoing epidemic?
English: line 194
Figure A, B: I suggest to use different colors. Figure b = What are the red dots and the black diamond symbols showing?
207: “The descriptive analysis of this study stipulated that the lowest temperature was 15.39 ⁰C whereas the highest was 31.86⁰C from January 1, 2020, to December 31, 2022, in Bangladesh”. Why is only covered the period between 2020 and 2022?, What about the previous years?
Line 208: “52636 paramount cases”. What do you mean by paramount?
Line 218: In the abstract, it is shown that the maximum daily confirmed cases is 52636. What is the period of time in which this number of cases was reported? Does this mean that in one day, the maximum number of reported cases was 52636?
Line 221-222: This phrase is repeated two times: “Correlations between meteorological factors and Dengue cases are shown in Figure 3.”
Table 1: The last line shows the “Daily confirmed cases (Number of persons)”, What is the number 52636 referring to? As it is the daily confirmed cases, this should reflect the number of dengue cases report in one day. The mean (831.33) should correspond to the average of the dengue cases report each day and the maximum would correspond to the maximum number of cases reported in one day.
-Figure 3: Legend of this figure explains that analysis correspond to data “from January 1, 2020, to December 31, 2022”, What happened with the previous years?
-Line 226: Figure 1? I believe that authors are referring to a different figure.
Line 247 and 255. Are you referring to Table 2?
Line 252-255: Why are you referring to monkeypox?
Check if it should be written dengue or Dengue?
Time series model results: Please clarify if these models were performed with data from 2000 to 2022 or to 2020 to 2022. In line 243 it is explained that “along with the confirmed and projected dengue cases from January 2020 to December 2022”.
Table 2: Why is it combined in the same table two and three columns?
Discussion
Line 368: I guess you mean Figure 2, instead of Figure 1.
Conclusion:
It should be more related to the results of the present study.
Please review the English and spelling of the manuscript:
Here I mention some examples:
Line 63: affiliated with the risk group
Line 297: English: This study has suspected that
Typo mistakes: 386 and other lines: oC
Line 396: Aedes in italics
Line 403: “It should be mentioned here that in the daytime, Aedes mosquitoes bite for that reason”
The manuscript is in the section of review, I think that the section research paper is better for it.
Author Response
RESPONSE TO REVIEWERS’COMMENTS
We take the pleasure to submit the revised manuscript (Manuscript ID: ijerph-2196638) entitled “An analysis of 23-years Dengue & meteorological correlation: persistent public health concern in Bangladesh” to be reconsidered for publication in renowned journal International Journal of Environmental Research and Public Health. The manuscript is thoroughly revised in accordance with the comments and suggestions made by the two reviewers.
Below is a detailed, point-by-point response addressing the specific comments and the reviewers’ concerns, including abstract, introduction, figures, tables, discussions, conclusions, and graphical abstract. All changes in the text are marked with GREEN. We have checked English grammar, edited all figures aesthetically, and analyzed data properly.
I do hope that all the issues and concerns raised by the reviewers have been addressed and that the revised manuscript will meet the standard of International Journal of Environmental Research and Public Health.
Comments and Suggestions for Authors
In this article six statistical models are applied to test the relationship between dengue-positive cases and meteorological parameters in Bangladesh. The idea of studying the association of dengue and meteorological variables, in a period of 23 years, is interesting. However, some parts of the manuscript are confusing. Please review the writing of the manuscript. Some sections of the manuscript (Introduction and conclusion) should be more directed to the relationship between dengue and meteorological variables. Another important point is that it is not clear the time period that was included in some of the analyses. It is explained that 23 years were covered in the study, but in some parts it is mentioned that data are from 2020 to 2022
Author's Response: We would like to thank the reviewer for identifying the important flaws in this manuscript. We edited the manuscript in the revised version as suggested.
Abstract:
Line 39: “During the study period, the mean daily DENV cases was 831.33±(3877.51), ranging between 0 to 52636”. As it is written it is understood that the maximum of daily dengue cases is 52636.
Author's response: The authors are grateful to the reviewer for his/her comments and corrected the line in the abstract.
Line 39: Not clear to which model(s) the authors are referring: “Though no substantial relationship exists between daily dengue cases and temperature, wind speed, and surface pressure.”
Author's response: The authors would like to acknowledge the reviewer for his/her valuable suggestion and we edited in the revised version.
-Why in the abstract only the results of the ARIMAX and GA models are shown?
Author's response: The authors would like to acknowledge the reviewer for his/her valuable question. We added others models in the revised manuscript.
-Line 43-45: It is mentioned the values of the ARIMAX and GA model, but is not clear if there is a relationship or not.
Author's response: We would like to thank the reviewer for identifying the important flaws and cleared in the revised version.
-Introduction:
The article is assessing the correlation between dengue and meteorological data. Complement the introduction with more information related to dengue and meteorological factors.
Author's response: The authors are thankful to the reviewer for his/her comments and added lines in the revised submitted manuscript.
It is not necessary to include the information that is given between line 64 and 70.
Figure 1 is not necessary.
Author's response: The authors are thankful to the reviewer for his/her comments and deleted the respective parts.
Results:
Line 193: It is not clear the period of time when the 126 dengue cases were recorded.
Author's response: The authors would like to thank the reviewer comment and edited the line.
Line 198: “Although the Dengue outbreak was tremendously devastating with 101354 patients in 2022”. It is not clear how this number is obtained. Figure 2A shows that in 2022 there are 62382 cases
Author's response: We appreciate the reviewer’s comments and edited the line in the revised manuscript.
Line 200: “the ongoing epidemic identified 16092 positive patients”. Not clear the period of time when these cases are recorded, what do you mean by the ongoing epidemic?
Author's response: The authors are thankful to the reviewer for his/her comments and edited the line.
English: line 194
Author's response: The authors are thankful to the reviewer for his/her comments and edited in revised version.
Figure A, B: I suggest to use different colors. Figure b = What are the red dots and the black diamond symbols showing?
Author's response: The authors are thankful to the reviewer for his/her comments and edited the figures.
207: “The descriptive analysis of this study stipulated that the lowest temperature was 15.39 ⁰C whereas the highest was 31.86⁰C from January 1, 2020, to December 31, 2022, in Bangladesh”. Why is only covered the period between 2020 and 2022?, What about the previous years?
Author's response: We acknowledge the reviewer’s thoughtful comment and edited it in the revised version.
Line 208: “52636 paramount cases”. What do you mean by paramount?
Author's Response: We would like to thanks the reviewer for identifying the important error and it is edited in the revised version.
Line 218: In the abstract, it is shown that the maximum daily confirmed cases is 52636. What is the period of time in which this number of cases was reported? Does this mean that in one day, the maximum number of reported cases was 52636?
Author's response: The authors are grateful to the reviewer for his/her comments and corrected it in the abstract.
Line 221-222: This phrase is repeated two times: “Correlations between meteorological factors and Dengue cases are shown in Figure 3.”
Author's response: The authors would like to acknowledge the reviewer for his/her valuable suggestion and we omitted the redundancy in revised version.
Table 1: The last line shows the “Daily confirmed cases (Number of persons)”, What is the number 52636 referring to? As it is the daily confirmed cases, this should reflect the number of dengue cases report in one day. The mean (831.33) should correspond to the average of the dengue cases report each day and the maximum would correspond to the maximum number of cases reported in one day.
Author's response: The authors would like to acknowledge the reviewer for his/her valuable suggestion. 52636 denote maximum daily confirmed cases of Dengue and average cases 831 are found.
-Figure 3: Legend of this figure explains that analysis correspond to data “from January 1, 2020, to December 31, 2022”, What happened with the previous years?
Author's response: We would like to thank the reviewer for identifying the important flaws and edited it.
-Line 226: Figure 1? I believe that authors are referring to a different figure.
Author's response: The authors are thankful to the reviewer for his/her comments and corrected it in the revised submitted manuscript.
Line 247 and 255. Are you referring to Table 2?
Author's response: The authors would like to acknowledge the reviewer for his/her valuable suggestion and we edited it in revised version.
Line 252-255: Why are you referring to monkeypox?
Author's response: The authors are grateful to the reviewer for his/her comments, and deleted.
Check if it should be written dengue or Dengue?
Author's response: The authors would like to acknowledge the reviewer for his/her valuable suggestion and we checked it in the revised version.
Time series model results: Please clarify if these models were performed with data from 2000 to 2022 or to 2020 to 2022. In line 243 it is explained that “along with the confirmed and projected dengue cases from January 2020 to December 2022”.
Author's response: We would like to thank the reviewer for identifying the important flaws and edited it.
Table 2: Why is it combined in the same table two and three columns?
Author's response: We would like to thank the reviewer for identifying the important flaws, three columns are used to indicate AIC, AICc, and BIC.
Discussion
Line 368: I guess you mean Figure 2, instead of Figure 1.
Author's response: The authors would like to acknowledge the reviewer for his/her valuable suggestion and we checked it in the revised version.
Conclusion:
It should be more related to the results of the present study.
Author's response: The authors are thankful to the reviewer for his/her comments and edited in revised submitted manuscript.
Please review the English and spelling of the manuscript:
Here I mention some examples:
Line 63: affiliated with the risk group
Author's response: We would like to thank the reviewer comments, and we edited in the revised manuscript.
Line 297: English: This study has suspected that
Author's response: The authors are thankful to the reviewer for his/her comments and corrected in revised submitted manuscript.
Typo mistakes: 386 and other lines: oC
Author's response: We would like to thank the reviewer for identifying the important flaws and edited in numerous places of revised manuscript.
Line 396: Aedes in italics
Author's response: The authors would like to acknowledge the reviewer suggestion and changed these manuscript criteria.
Line 403: “It should be mentioned here that in the daytime, Aedes mosquitoes bite for that reason”
Author's response: The authors would like to acknowledge the reviewer for his/her valuable suggestion and we edited in the revised version.
The manuscript is in the section of review, I think that the section research paper is better for it.
Author's response: The authors would like to acknowledge the reviewer suggestion, and we will change these manuscript criteria.

Reviewer 2 Report
The article "An analysis of 23-years Dengue & meteorological correlation: 2 persistent public health concern in Bangladesh", title is unclear. What is 23-years Dengue…. ? It does not give a proper sense, need to rephrase it.
The autrhor's contribution is not properly defined. “MAI has conceived and designed the study. All authors read and approved the final draft of the manuscript.” Write explicitly.
Below are the comments and suggestions:
· Do not use abbreviations in the abstract and main text without explaining these.
· There are some unnecessary long sentences; avoid these.
· The below sentences do not make any proper sense here. Do you want to write about COVID-19 or DENV?
In 2026, Bangladesh, which is currently developing, will reach middle-income status. Dealing with coronavirus was one of the most challenging issues for 88 Bangladesh. Although the Government of Bangladesh (GoB) has tackled this situation, daily COVID-19 cases are still being recorded.
· Equation 2 is not referred to anywhere in the main text.
· In line 181, write the equation number and refer to the main text.
· Clarify the below sentences in line number 221 and 222.
· Correlations between meteorological parameters and Dengue cases are presented in Figure 3. Correlations between meteorological factors and Dengue cases are shown in Figure 3.
· Highlight the significance of the study.
· Re-write the conclusion section in a concise way, include some considered results too.
Author Response
RESPONSE TO REVIEWERS’COMMENTS
We take the pleasure to submit the revised manuscript (Manuscript ID: ijerph-2196638) entitled “An analysis of 23-years Dengue & meteorological correlation: persistent public health concern in Bangladesh” to be reconsidered for publication in renowned journal International Journal of Environmental Research and Public Health. The manuscript is thoroughly revised in accordance with the comments and suggestions made by the two reviewers.
Below is a detailed, point-by-point response addressing the specific comments and the reviewers’ concerns, including abstract, introduction, figures, tables, discussions, conclusions, and graphical abstract. All changes in the text are marked with GREEN. We have checked English grammar, edited all figures aesthetically, and analyzed data properly.
I do hope that all the issues and concerns raised by the reviewers have been addressed and that the revised manuscript will meet the standard of International Journal of Environmental Research and Public Health.
Comments and Suggestions for Authors
The article "An analysis of 23-years Dengue & meteorological correlation: persistent public health concern in Bangladesh", title is unclear. What is 23-years Dengue…. ? It does not give a proper sense, need to rephrase it.
Author's response: The authors would like to acknowledge the reviewer for his/her valuable suggestion and we edited title in the revised version.
The author's contribution is not properly defined. “MAI has conceived and designed the study. All authors read and approved the final draft of the manuscript.” Write explicitly.
Author's response: The authors are thankful to the reviewer for his/her comments and corrected in revised submitted manuscript.
Below are the comments and suggestions:
Do not use abbreviations in the abstract and main text without explaining these.
Author's response: We would like to thank the reviewer for identifying the important flaws and edited in numerous places of revised manuscript.
There are some unnecessary long sentences; avoid these.
Author's response: The authors are thankful to the reviewer for his/her comments and corrected in revised submitted manuscript.
The below sentences do not make any proper sense here. Do you want to write about COVID-19 or DENV?
In 2026, Bangladesh, which is currently developing, will reach middle-income status. Dealing with coronavirus was one of the most challenging issues for 88 Bangladesh. Although the Government of Bangladesh (GoB) has tackled this situation, daily COVID-19 cases are still being recorded.
Author's response: We would like to thank the reviewer for identifying the important flaws and deleted in the revised manuscript.
Equation 2 is not referred to anywhere in the main text.
Author's response: The authors would like to acknowledge the reviewer suggestion and added in the revised manuscript.
In line 181, write the equation number and refer to the main text.
Author's response: The authors would like to acknowledge the reviewer suggestion and added in the revised manuscript.
Clarify the below sentences in line number 221 and 222.
Correlations between meteorological parameters and Dengue cases are presented in Figure 3. Correlations between meteorological factors and Dengue cases are shown in Figure 3.
Author's response: The authors would like to acknowledge the reviewer for his/her valuable suggestion and we edited in the revised version.
Highlight the significance of the study.
Author's response: The authors would like to acknowledge the reviewer suggestion and added in the revised manuscript.
Re-write the conclusion section in a concise way, include some considered results too.
Author's response: The authors would like to acknowledge the reviewer suggestion and added in the revised manuscript.

Round 2
Reviewer 2 Report
The authors have incorporated the required changes. I do not have further comments.